# Recent Advances in Transition Metal Phosphide Nanocatalysts for H$_2$ Evolution and CO$_2$ Reduction

**Saman Shaheen, Syed Asim Ali, Umar Farooq Mir, Iqra Sadiq and Tokeer Ahmad *** 

Nanochemistry Laboratory, Department of Chemistry, Jamia Millia Islamia, New Delhi 110025, India;
samanshaheen@gmail.com (S.S.); syedasimali954@gmail.com (S.A.A.); umarmir310@gmail.com (U.F.M.);
iqrasadiq62@gmail.com (I.S.)
* Correspondence: tahmad3@jmi.ac.in; Tel.: +91-11-26981717 (ext. 3261)

**Abstract:** Green hydrogen energy has captivated researchers and is regarded as a feasible option for future energy-related aspirations. The emerging awareness of renewable energy-driven hydrogen generation and carbon dioxide reduction calls for the use of unconventional schematic tools in the fabrication of nanocatalyst systems. Transition metal phosphides are state-of-art, cost-effective, noble-metal-free materials that have been comprehensively examined for sustainable energy-driven applications. Recent reports on these advanced functional materials have cemented their candidature as high-performance catalytic systems for hydrogen production and for carbon dioxide conversion into value-added chemical feedstock. Bimetallic NiCoP (238.2 mmol g$^{-1}$ h$^{-1}$) exhibits top-notch catalytic competence toward photocatalytic HER that reveals the energy-driven application of a pristine class of TMPs, whereas heterostructured Ni$_2$P/CdS was found to be fit for photochemical CO$_2$ reduction, as well as for HER. On the other hand, pristine Ni$_2$P was recently ascertained as an efficient electrocatalytic system for HER and CO$_2$RR applications. A wide array of physico-chemical modulations, such as compositional and structural engineering, defect generation, and facet control, have been used for improving the catalytic efficiency of transition metal phosphide nanostructures. In this review, we succinctly discuss the proficiency of transition metal phosphides in green hydrogen production and carbon dioxide conversion via photochemical and electrochemical pathways. We detail the significance of their structural properties and brief the readers about the synthetic advancements without deviating from our goal of summarizing the recent achievements in energy-driven applications.

**Keywords:** transition metal phosphides; CO$_2$ reduction; H$_2$ generation; photocatalysis; electrocatalysis

## 1. Introduction

The devastating effects of global warming and climate change have motivated researchers to focus their attention beyond fossil fuels and their various modified derivatives. The unchecked release of carbon dioxide (CO$_2$) has drastically perturbed the flora and fauna of Earth's ecosystems and has put several species on the verge of extinction, as well as threatened the adaptation of many others [1]. Annually, 32.8 billion tonnes of CO$_2$ are released into the atmosphere from the consumption of fossil fuel derivatives such as coal and petroleum [2]. According to the special report proposed by the International Energy Agency (IEA), it is anticipated that the global energy demand will rise by about 25% in the coming two decades because of population expansion, and the energy associated with CO$_2$ release will increase by about 35.8 megatons per year in 2040 [3]. These catastrophic predictions and current environmental maladies necessitate a paradigm shift to a sustainable energy solution. Various accords such as the Kyoto Protocol and the Paris Agreement have been signed to combat CO$_2$ release collectively at multiple levels [4]. The inherent heating nature of CO$_2$ has the potential to cause fluctuations in the seasonal patterns of El Nino and La Nina, which govern the nature of winds worldwide [5]. If standard protocols are

not implemented to control the unrestricted venting of $CO_2$, the atmosphere of our planet will be like that of Venus, and the planet will become a non-colonized gas chamber [6]. Consequently, the average global temperature and sea level will increase.

The development of any nation depends heavily on the manufacturing, automotive, and agriculture sectors, but all of these industries require efficient fuels for consequential output. Hence, the optimization of energy resources is economically fundamental for any nation to achieve an equilibrium between the demand and supply chains. Hydrogen ($H_2$), as a renewable energy resource, has various benefits such as remarkable gravimetric density and higher calorific value, which make it a viable alternative to fossil derivatives [7–10]. Currently, steam reforming of fossil fuels is the standard practice for the generation of bulk $H_2$. However, the nature of this type of $H_2$ is not at all eco-friendly and augments greenhouse gas emission. Therefore, green $H_2$ energy via overall water splitting is sought out as an environmentally benign way to extract $H_2$ [11]. Currently, the research community is committed to elevating the efficiency of the $H_2$ evolution reaction (HER) and $CO_2$ reduction reaction ($CO_2$RR) to the extent that they become a tandem antidote to the climatic concerns of the planet.

The development of HER and $CO_2$RR is hindered by the non-achievement of cost-effectiveness and efficiency in meeting large-scale operational endeavors. HER and the conversion of $CO_2$ into value-added chemicals such as hydrocarbon fuels via photochemical, electrochemical, and photo-electrochemical pathways offer a great deal of cost efficiency and production output because these green routes have been significantly scrutinized [12–16]. Nanocatalysis has proven to be an effective pathway to an environmentally benevolent protocol for $CO_2$ reduction and $H_2$ generation [17]. Nanostructures exhibit significantly higher exposed active sites, and because of quantum confinement effects, they possess optoelectronic properties that are superior to their bulk counterparts [18]. Therefore, exploiting nanocatalysts to carry out HER and $CO_2$RR is instrumental to a scalable production of $H_2$ and carbon-based value-added chemicals. There are numerous fabrication techniques that have been explored for designing the nanostructures of advanced functional materials, such as the reverse micellar system [19–22], the polymeric precursor route [23], the hydrothermal/solvothermal method [24], etc.

$CO_2$ is an ideal starting material for the production of energetic chemical fuels that are desired to meet energy requirements in the future [25]. Photochemical and electrochemical $CO_2$RR pathways are considered to be economical and environmentally benign routes for the conversion of $CO_2$ into valuable feedstocks such as $CO$, $HCOOH$, $CH_4$, $C_2H_4$, $CH_3COCH_3$, and $CH_3OH$, without producing any undesirable by-products [26–28]. The schematic representation of the photocatalytic conversion of $CO_2$ into valuable feedstock is demonstrated in Figure 1. In addition, there is a process of syngas ($CO+H_2$) generation via the photocatalytic/electrocatalytic reduction of $CO_2$ in an aqueous solution that is a vital intermediate for enhancing the yield of hydrocarbon fuels. The reverse water–gas shift reaction (RWGS) is a sustainable pathway of $CO_2$ conversion that has received recognition for its widespread application of $CO_2$ as the starting material [29]. However, all of these HER and $CO_2$RR routes demand Earth-abundant, advanced functional materials that can throttle secondary reactions such as the back recombination of charge carriers, the methanation reaction, or competing HER reactions in the case of $CO_2$ sequestration.

Various classes of materials have been explored as electrocatalysts and photocatalysts, including metals [30], metal alloys [31], metal oxides [32,33], metal sulfides [34], metal-organic frameworks (MOFs), transition metal chalcogenides [35,36], and carbon-based materials like graphene or carbon nanotubes [37,38]. However, metal oxides exhibit low conductivity, and metal sulfides are limited by their photo-corrosiveness. Transition metal phosphides (TMPs) have emerged as efficient catalysts with high electrical conductivity and exceptional physicochemical properties. The catalytic efficiency of TMPs is highly dominated by their intrinsic properties such as light absorptivity, the presence of active sites for the adsorption of reactive species, transportation capability, tunable band potential, and photosensitization. It is relatively difficult to obtain scalable energy-related performance via

pristine TMPs, so in order to achieve higher performance, various strategies have been formulated, such as synthetic modulations, cocatalyst incorporation, heterojunction formation, and compositional and morphological modifications. The fabrication of TMPs fundamentally hinges upon the metal-to-phosphorus ratio and the sources of phosphorus. Therefore, researchers are trying to optimize the synthetic methodologies of TMPs that elute non-toxic or less harmful by-products and that are environment friendly in nature. Although TMPs exhibit higher catalytic response for HER and $CO_2RR$ applications, there are still several bottlenecks that need to be rectified prior to their large-scale commercial application.

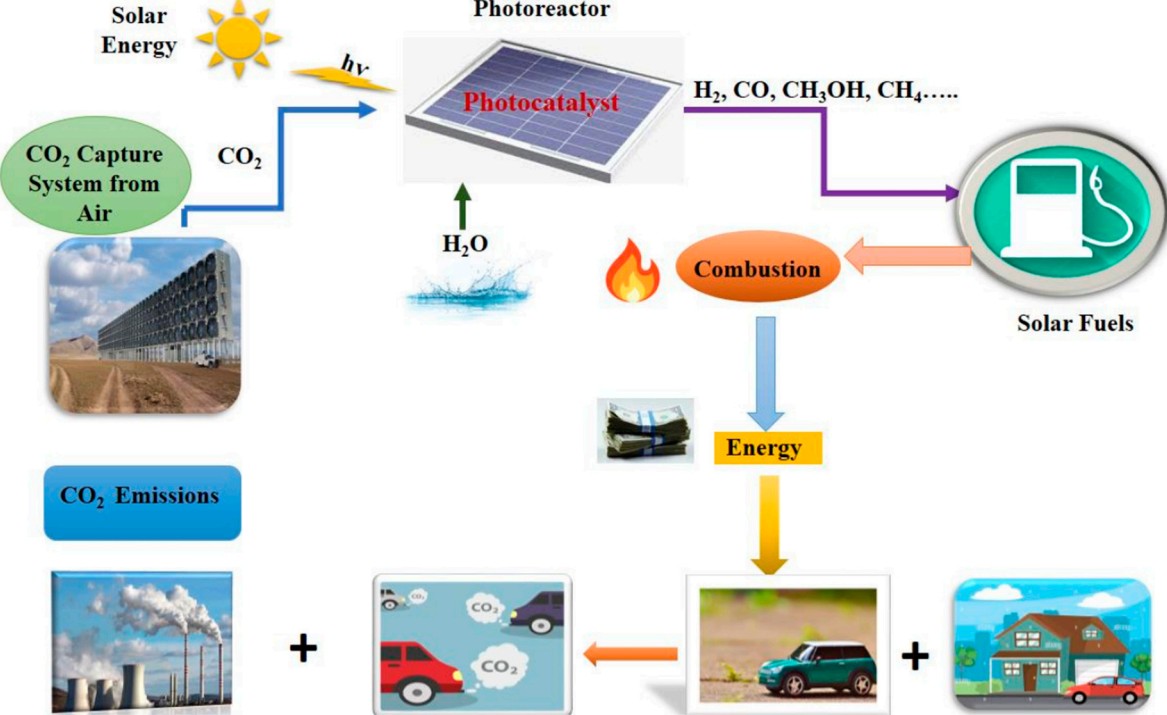

**Figure 1.** Schematic demonstration of photocatalytic reduction/conversion of $CO_2$ into solar fuel. [Reprinted with permission from Ref. [1]. Copyright 2023, John Wiley and Sons].

TMPs are an emerging class of advanced materials believed to be potent candidates that can replace precious noble metal-based catalysts [39]. The role of TMPs in nanocatalysis is not only limited to direct utilization. Instead, they can also be employed as cocatalysts. In the last few decades, it has been observed that cocatalyst loading is one of the best ways to enhance photochemical and electrochemical efficiencies by improving their light-harvesting, electrical conductivity, and physicochemical properties through interface engineering, alongside providing multifunctional active sites for surface adsorption [40,41]. Both the activity and selectivity of HER and $CO_2RR$ can be enhanced by using cocatalysts. Among different types of cocatalysts, noble metals such as Pt, Pd, Au, Ag, and Ru have been comprehensively studied for the photocatalysis of HER and $CO_2RR$ [42–45]. However, due to their scarcity and the whopping cost involved, their large-scale applicability is limited. An advanced, economical, and efficient method is the need of the hour. Therefore, developing non-noble-metal-based cocatalysts which fulfil the requirement of being cost-effective is requisite. Among various non-noble-metal-based cocatalysts, TMPs are classified as promising candidates for photocatalysis and electrocatalysis. Antil et al. [44] fabricated novel NiCoP@ZnCo-MOF photocatalysts and displayed an augmented rate of $H_2$ (8583.4 µmol g$^{-1}$ h$^{-1}$). The group conducted comparative analysis between as-synthesized TMP nanocatalysts relative to the noble-metal-based catalytic system of Pt@ZnCo-MOF (8885.7 µmol g$^{-1}$ h$^{-1}$). The rate of $H_2$ production of the TMP-based photocatalyst was found to be nearly analogous to its noble metal counterpart. Thus, we can corroborate

TMPs as a viable substitute for noble metals in assisted catalysis for advanced HER applications. An illustrated diagrammatical comparison is provided in Figure 2 for the better selectivity of TMPs over noble metal catalysts.

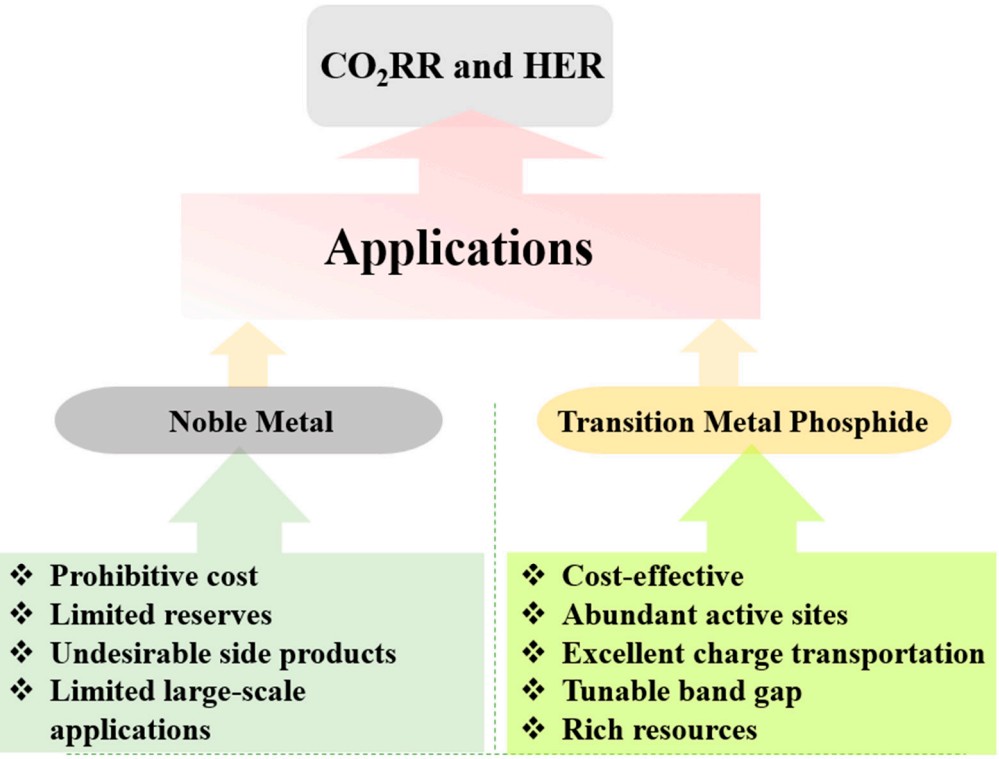

**Figure 2.** Diagrammatic illustration of $CO_2RR$ and HER with respect to noble metal and transition metal phosphide.

The schematic landscape of TMPs synthesis, properties, and energy-driven applications such as energy conversion and energy storage are illustrated in Figure 3. In 2005, for the very first time, Liu et al. [46] predicted the catalytic excellence of $Ni_2P$ by considering density functional theory and evaluating its photocatalytic activity towards HER. The scope of the CoP cocatalyst has been analyzed to augment the photocatalytic efficiency of two-dimensional $g$-$C_3N_4$. In the CoP/$g$-$C_3N_4$ heterostructured catalyst, photo-generated electron transference occurs from $g$-$C_3N_4$ to the surface of CoP active sites in light [46,47]. The surface electrons react with the absorbed protons, resulting in the supplication of photocatalytic HER activity. Recent reports have also proven the potential of other nickel-based TMPs such as $Ni_2P$, $Ni_{12}P_5$, and $Ni_3P$ for enhancing the light absorption capacity of $g$-$C_3N_4$, due to which they are believed to possess high photocatalytic activity in HER [48]. TMPs have achieved superior photocatalytic HER and $CO_2RR$ activities, as revealed by the literature review [49]. Hence, researchers need to focus on the developments in the field of TMP-assisted nanocatalysis. This review summarizes the recent advances in TMPs towards HER and $CO_2RR$ applications.

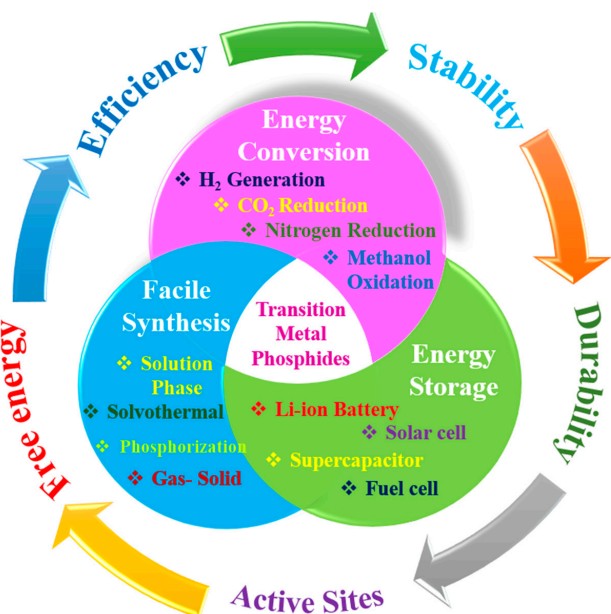

**Figure 3.** Schematic landscape of TMPs synthesis, properties, and energy driven applications.

## 2. TMPs Structure and Its Significance

TMPs are versatile catalysts that consist of transition metals bonded with phosphorus (P) atoms. The structure of TMPs is of great significance and can vary depending on the types of metals, the number of P atoms, and the bonding interactions between them. TMPs possess different structures and morphology depending on the other metals and their interactions with the P atoms [50]. In addition, factors like the reactant precursors of transition metal and P, stoichiometry, and synthesis conditions also govern the structural properties of TMPs [51]. In TMPs, the P atom enters the lattice structure of transition metals to form interstitial compounds that act as efficient catalysts for energy conversion processes [52]. The structural modifications on introducing the P atom in the lattice structure of transition metal (M) is an elongation in the bond length of the M-P bond, which results in decreased interactions of M-M bonds and the compression of d-bands [53]. This lattice mismatch tunes the Fermi levels of M, favoring their electronic conductivity and mass diffusion characteristics [54]. TMPs exhibit unique crystal structure where the M atoms form the trigonal prismatic structure, in which the interstitial voids are occupied by P atoms in the smallest structural unit. In contrast, the excess metal ions form nine-fold tetrakaidecahedral structures [55], as represented in Figure 4a,b. The P atom can interact with the transition metal lattice. These structural units can cooperate and have the ability to form miscellaneous lattice types. Variations in the M/P ratio significantly affect the lattice structure of TMPs. Thus, this ratio also controls the catalytic efficiency of TMP nanocatalysts. In a TMP structure, P exhibits relatively higher electronegativity, attributed to the fact it accepts protons readily. This structural feature of TMPs boosts their catalytic action towards HER and $CO_2$RR as facilitated $H^+$/$CO_2$ surface adsorption takes place quickly due to higher electronegativity of P atoms [50]. Therefore, the P content in TMPs is central to determining their catalytic activity. Based on that, they can be classified as transition-metal-rich TMPs (M-rich TMPs) or P-atom-rich TMPs (P-rich TMPs), depending on the M/P ratio [56]. In M-rich TMPs, the M/P ratio is comparatively higher, due to which M–M interactions profoundly affect the physiochemical properties of TMPs. These compounds often exhibit a lower P content than stoichiometric or P-rich TMPs [57]. The excess of M atoms results in unique properties and structures of TMPs. M-rich TMPs are commonly synthesized under high-temperature conditions, such as through the reaction of metal precursors with P sources. However, P-rich TMPs hold a relatively higher ratio of P to transition metal. These compounds have excess P and exhibit unique properties based on their high P content. The crystal structure of P-rich TMPs is depicted in Figure 5a–f below [56]. M-rich TMPs

exhibit metallic properties, whereas P-rich TMPs show semiconducting behavior. The metallic behavior in M-rich TMPs is familiar with that of noble metals, showing excellent conducting properties [42]. P-rich TMPs are often synthesized through reactions that involve the addition of excess P or P-rich precursors. For instance, M-rich TMPs bestow higher conductivity and stability than P-rich TMPs due to the larger M–M and M–P bonds than non-conductive P–P bonds.

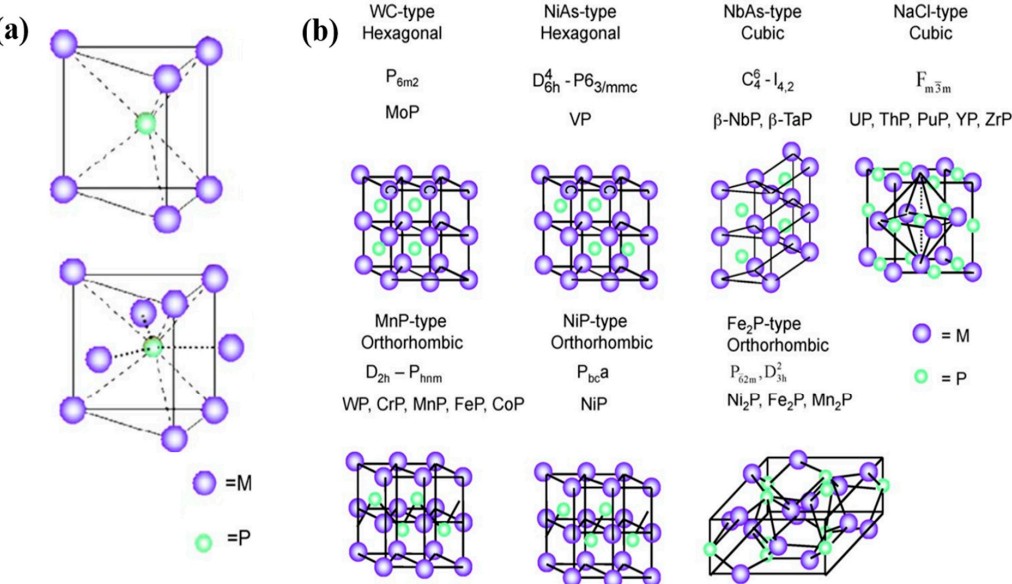

**Figure 4.** (**a**) Triangular prism and tetrakaidecahedral structures in phosphides and (**b**) crystal structures of metal-rich phosphides. [Reprinted with permission from Ref. [55]. Copyright 2009, Catalysis].

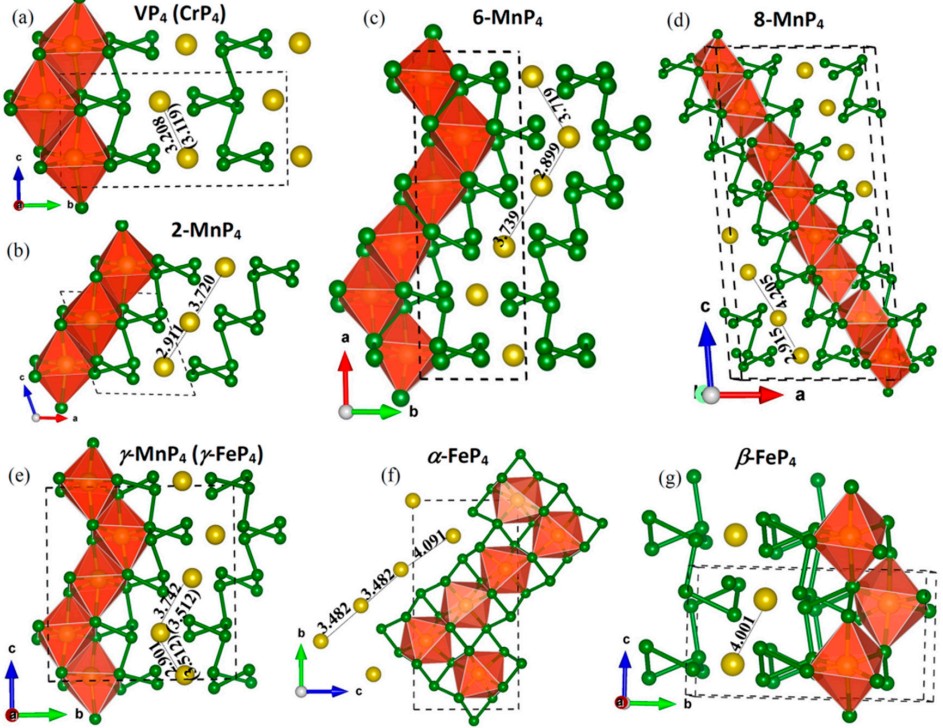

**Figure 5.** Crystal structures of 3D-TMP$_4$ (TM = V, Cr, Mn, and Fe) polyhedron view. The TM and P atoms are represented as big yellow spheres and small green spheres, respectively. [Reprinted with permission from Ref. [56]. Copyright 2018, American Chemical Society].

## 3. Structural Significance

The structural ascendency of TMPs, due to the introduction of P-atoms in the lattices of transition metal ions, dramatically affects the optoelectronic properties of these interstitial materials, thus making them excellent HER and $CO_2$RR catalysts [58–60]. Incorporated P atoms moderately tune the interspace between the transition metal atoms that restricts their molecular interaction. It leads to the shrinking of d bands and increasing the concentration of energy levels in the vicinity of the Fermi center [61]. This excellent optoelectronic modulation in TMPs escalates their catalytic efficiency of the order of noble-metal based catalytic systems [62,63]. The structural engineering of TMPs is one of the finest strategic tools to ameliorate their performance towards HER and $CO_2$RR applications. Recent reports of TMPs support the fact that developing heterojunctions, hollow and nanoarray structures, significantly enhance their scope of applicability. Heterojunction formation allows for the interfacial charge transfer during photochemical HER and $CO_2$RR, which effectively subdues the back recombination reactions and assures the utilization of active electron carriers in primary reduction reactions. On the other hand, the formation of hollow structures eases the adsorption pathways of reactant species by providing higher exposed active sites available for surface reactions and sufficient mass diffusion routes [64–68]. Higher surface areas and surface roughnesses in nanoarrays facilitate the contact compass between catalytic system and adsorbed species. The structural properties of TMPs are pretty significant in modulating their electronic properties, leading to remarkable catalytic responses in energy-related applications. Other than structural engineering, tuning the physicochemical properties of TMPs also hinges on their compositional and morphological properties.

## 4. Advances in TMP Fabrication

For the comprehensive exploitation of TMPs, optimizing synthetic routes is essential to achieve cost-effectiveness and environmental friendliness. Conventional methods used to fabricate TMPs, such as temperature-programed reduction under $H_2$ atmosphere or the ball milling method, are restricted to the synthesis of bulk TMPs only. However, for constructing nano-dimensional TMPs, fabrication routes can be classified on the basis of P sources used and the different methods involved. Synthetic pathways have been classified based on P sources such as organic phosphines (trioctylphosphine), hypophosphites ($NH_4H_2PO_2$, $NaH_2PO_2$), elemental phosphorus (white, red), and phosphorenes [13]. Phosphorenes are the exfoliated monolayers of black P that have also been utilized as P precursors for TMP engineering [69]. One different approach to designing TMPs is the pyrolysis of P and carbon-based precursors alongside the source of the transition metal ion. Integrating carbon-based materials during TMP synthesis impedes the agglomeration of TMP nanoparticles and accelerates their conductivity. Therefore, different morphologies and particle sizes can be achieved based on the use of varying P sources for TMP nanostructures. The discrete synthesis method for designing TMPs can vary metal to metal or depending on the desired morphology, the targeted energy-related application, and the chosen precursors of P [70]. Researchers often optimize the reaction conditions and other parameters such as reducing the agent and solvent system to obtain the desired physicochemical properties and morphology of the prepared TMP nanocatalysts [71]. Synthesizing TMP nanostructures typically involves diverse physical and chemical routes such as solid-state reactions [72], the phosphorization of metal precursors [73–76], chemical vapor deposition (CVD) [77], and hydrothermal pathways [78–82].

In the usual practice, the solid-state method proceeds via heating a mixture of transition metal precursors (metallic powder or metal oxides) alongside a P source at significantly higher temperatures under an inertial atmosphere to subdue any undesired defects or oxide impurities [72]. The reaction advances through a series of intermediate steps that lead to the development of the desired TMP nanostructures. As its name suggests, the phosphorization of metal precursors comprises the precursor of the desired metal and is minted with the source of P, such as red or white phosphorus. The phosphorization reaction can occur either in solution or in the gas phase, depending on the nature of transition metal precursor and

P source. The obtained reaction mixture is then further processed to isolate the pristine TMPs without any foreign impurities. The CVD technique is generally utilized to deposit thin films or coatings of TMPs. In this pathway, relatively volatile metallic precursor is integrated with the P-containing precursor, and the resulting mixture is introduced into the reaction chamber, followed by that reactant mixture decomposed at high temperatures leading to the deposition of the TMPs onto a substrate. The hydrothermal/solvothermal route is an environmentally benign chemical route that offers great deal of output and control over morphology for far-reaching applications of TMP nanostructures [80]. It involves the reaction of metal precursors with the P source in generally aqueous media and sometimes in other solvents under high-pressure and high-temperature conditions. The organic solvent environment is developed in the reaction media to achieve the desired morphology. The reaction is typically carried out in an autoclave, where the reaction mixture is subjected to the particular reaction conditions. The main advantage of procuring this route is the formation of well-defined TMP nanoparticles. Considering the aforementioned synthetic routes of TMPs, we manifest that all these pathways have their respective advantages and disadvantages. However, in our vision, the hydrothermal route has far-reaching applications for the sake of energy-related operations of TMP-based hybrid materials.

## 5. TMP Photocatalysts for HER and $CO_2$RR Applications

Over the past few years, the optoelectronic properties and chemical stability of TMPs have made them promising catalytic systems for photochemical HER and $CO_2$RR applications. TMPs such as FeP [83], CoP [84], NiCoP [85], MoP [86], and $Cu_3P$ [87] have been investigated in the field of nanocatalysis and are known to display exceptional activity and selectivity towards HER and $CO_2$RR applications. There are various methods of enhancing the photocatalytic performances of TMPs such as metallic/non-metallic doping [88], surface modulation [89], and mixing with other nanomaterials to form heterojunctions [90]. For instance, the integration of noble metals like Au and Pt with TMPs ameliorates their photocatalytic activity towards HER and $CO_2$RR [49,91]. In the past few decades, much of the attention has been placed on improving the efficiency of photocatalytic processes by developing and probing miscellaneous catalytic systems. Therefore, to achieve this goal, researchers are persistently trying to enhance the activity and selectivity of TMP-based photocatalysts. However, this transformation is limited by the quick recombination rate of photogenerated electron–hole pairs and the restricted photo-sensitization of light absorbers during photocatalytic operations. These two obstacles are the major bottlenecks of photocatalytic HER and $CO_2$RR operations that impede their quantum efficiency in achieving $H_2$ and carbonaceous value-added chemicals [49]. The tailored band energy of TMP nanocatalysts allows an accelerated harvesting of light that ultimately leads to the flow of uninterrupted photo-induced charge carriers, allowing longer runs of photocatalysis. The prevention of the back recombination of photo-generated electron–hole pairs can be successfully achieved by developing appropriate cocatalysts that would generate suitable interfaces, such as Schottky junctions, S-schemes, Z-schemes, p-n heterojunctions, and type II heterojunctions, leading to the fast movement of charge carriers from one component to the other, thus reducing the probability of electron–hole pair recombination [92–94]. Although, some secondary side reactions result in the photocatalytic $CO_2$RR. HER is one of the competent reactions in the conversion of $CO_2$ to CO. Therefore, the recent trends of TMP-assisted photocatalytic applications have imbibed these limiting factors, as demonstrated by the reports of TMPs discussed hereafter in this section.

Li et al. [78] fabricated a $Ni_2P/ZnIn_2S_4$ heterostructure to examine its scope towards photocatalytic water splitting. Optimized $Ni_2P/ZnIn_2S_4$ heterojunctions exhibited a remarkable $H_2$ evolution rate of ~2.1 mmol $g^{-1}$ $h^{-1}$, which can be attributed to the better synergism between the photocatalytic components, the enhanced charge separability, the larger surface area, and the outstanding electron transport ability. Shen et al. [95] synthesized EosinY-$Cu_3P$-CNT (carbon nanotubes). The group reported an exceptional $H_2$ evolution rate of 17.22 mmol $g^{-1}$ $h^{-1}$, which is owed to the efficient separation of charge

carriers in Eosin Y and enhanced electron transfer ability that symbiotically boosted the rate of $H_2$ generation in the TMP-based photocatalyst. Meng et al. [40] fabricated Ru-CoP-1:8/GCN (g-$C_3N_4$) photocatalysts via the wet chemical reduction method for efficient $H_2$ production. The reported study showed an exceptional rate of 1172.5 $\mu$mol g$^{-1}$ h$^{-1}$. This report outlines the cost-effectiveness of Ru-CoP heterostructured catalytic system as compared to noble metal catalysts for photocatalytic applications. Gong et al. [86] developed Ni-MoP@NP$_{PF}$ (porous nitrogen-doped carbon nanofibers) via electrospinning followed by phosphorization as well as carbonization and showed the rate of CO production to be 953.53 $\mu$mol g$^{-1}$ h$^{-1}$. It exhibited an impressive selectivity for CO of 98.95% due to the formation of Ni-N bonds. The mechanism proposed that the transfer of electrons occurs across the interface between the two catalysts, enhancing the rate of $CO_2$ reduction, as represented in Figure 6a. The photocatalyst revealed exceptional activity and selectivity for $CO_2$ reduction with Ni, as without Ni co-catalyst, the yield was found to be relatively inferior, as depicted in Figure 6b. The product selectivity of Ni-MoP@NC$_{PF}$ is realized by higher CO generation, as the optimized Ni-MoP@NC$_{PF}$ catalytic system exhibited greatly enhanced photocatalytic $CO_2$ reduction performance towards CO generation compared to $CH_4$ and $H_2$ product formation, as depicted in Figure 6c. Ni-MoP@NC$_{PF}$ provided channels for $CO_2$ diffusion and electron transport. This type of TMP-based catalytic system is the epitome of collaborative morphological, compositional, and hetero-interfacial engineering, as the distribution of Ni throughout the MoP@NC$_{PF}$ improved the transformation of $CO_2$ to CO. The main factors that regulate the degree of photocatalytic HER and $CO_2$RR processes are the separation, transfer, and recombination of electron–hole pairs. The photocatalytic activity of $CO_2$ conversion is ameliorated by the higher separation rate of photo-induced charge carriers or lower recombination rate ascribed to more electrons available for primary reduction reactions at the conduction band of TMP-based photocatalysts.

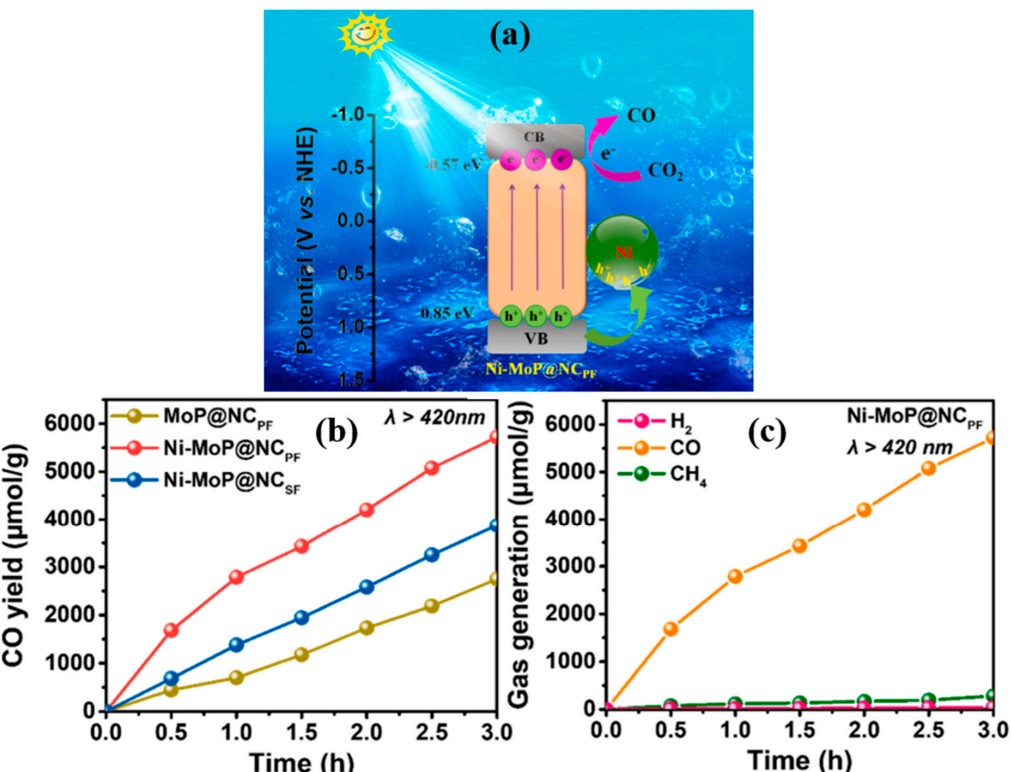

**Figure 6.** (**a**) The mechanism of Ni-MoP@NCPF as a photocatalyst for $CO_2$ reduction, (**b**) CO generation on various photocatalysts under visible light irradiation (with a UVCUT-420 nm filter) for 3 h, and (**c**) gas products of $CO_2$ reduction on Ni-MoP@NC$_{PF}$ under visible light irradiation for 3 h. [Reprinted with permission from Ref. [86]. Copyright 2022, Elsevier].

Sun et al. [79] fabricated $Cu_3P$-$Ni_2P$/g-$C_3N_4$ nanocatalysts via a solvothermal pathway to determine their photochemical HER efficiency. The $H_2$ production rate was found to be considerably efficient due to the formation of p-n heterojunctions, assisting in the facile charge transfer and exposure of a greater number of active sites. Figure 7a,b contain mechanistic sketches of $Cu_3P$-$Ni_2P$/g-$C_3N_4$-assisted photocatalytic $H_2$ generation and the transfer of electrons via heterojunctions after the advent of an internal electric field before and after visible light irradiation, respectively. Song et al. [96] fabricated WP-NC (nitrogen-doped carbon)/g-$C_3N_4$ heterojunctions for efficient $H_2$ generation via photocatalytic and electrocatalytic pathways. The optimized WP-NC/g-$C_3N_4$ catalyst exhibited an $H_2$ evolution rate as high as 1.2 mmol $g^{-1}$ $h^{-1}$, owing to the deposition of NC layers over WP, which ameliorated the charge separation and transfer efficiency during photocatalytic water splitting.

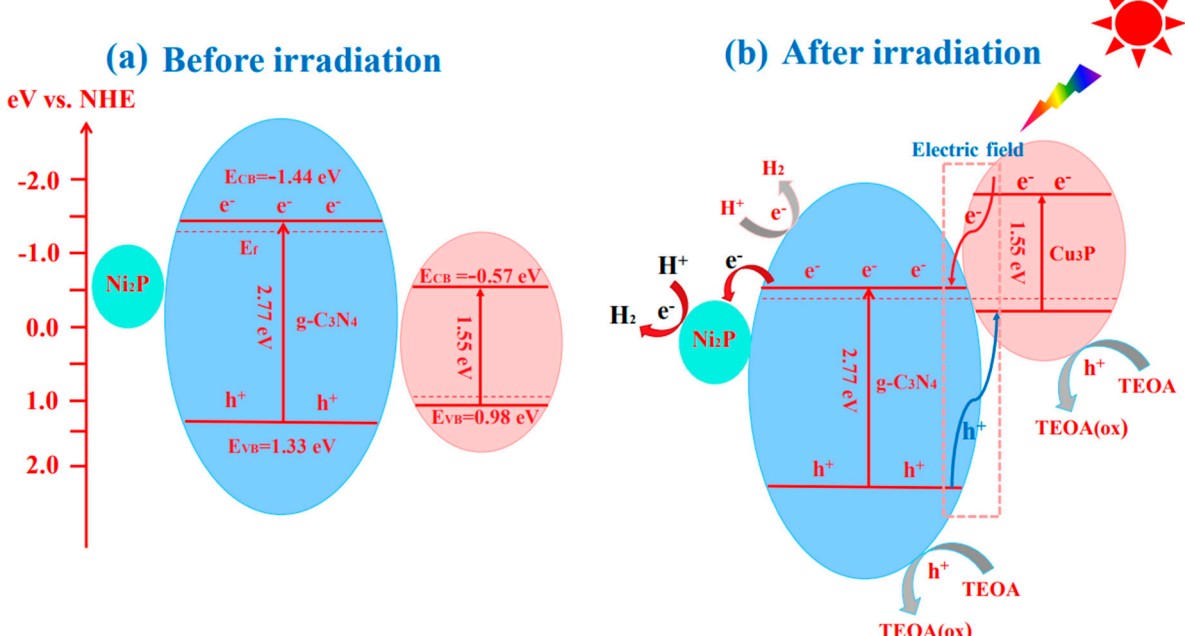

**Figure 7.** Mechanism of photocatalytic $H_2$ production for $Cu_3P$-$Ni_2P$/g-$C_3N_4$. [Reprinted with permission from Ref. [79]. Copyright 2020, John Wiley and Sons].

Su et al. [80] engineered $Ni_2P$/CdS photocatalysts via the hydrothermal pathway and studied their efficiency for HER and $CO_2$RR applications. The resultant band gap of the $Ni_2P$/CdS photocatalyst was found to be 2.16 eV with multiple production rate of $H_2$, CO, and $CH_4$ generation with porous and non-porous nanocomposites, as depicted in Figure 8. In this report, heterojunction formation played an essential role in the enhanced photocatalytic efficiency of $Ni_2P$/CdS, ascribed to the facilitated charge transfer. Guo et al. [83] synthesized an FeP/CN photocatalyst via the thermal decomposition method and reported the maximum rate of CO production to be 5.19 µmol $g^{-1}$ $h^{-1}$. The synthesized catalysts exhibited the tremendous conversion efficiency of $CO_2$ to CO because of the greater active sites and facile charge transfer during the photocatalytic $CO_2$RR. Wang et al. [97] designed and synthesized novel Fe-doped CoP for $CO_2$ reduction to CO. The results reported a maximum selectivity of 90.3% (CO), and the rate of evolution was estimated to be 21 µmol $h^{-1}$. Fe doping assisted in subduing the activation energy barrier for intermediate formation and promoted CO evolution. Lv et al. [98] synthesized ultrathin NiCoP nanosheets to achieve an effective rate of HER (238.2 mmol $g^{-1}$ $h^{-1}$). The reported bimetallic catalysts exhibited bifunctional properties such as electrocatalysis and photocatalysis efficiently. Duo et al. [99] designed FeP/CdS heterostructured photocatalysts via the solvothermal route, and the as-prepared nanocatalyst produced $H_2$ at the rate of 37.92 mmol $g^{-1}$ $h^{-1}$.

This excellent HER was attributed to the charge separation and better electron transportation demonstrated by the heterojunction formation between the FeP and CdS. Li et al. [72] fabricated $Ni_2P/NiO/CN$ (graphitic carbon nitride) via solid–gas reaction for the conversion of $CO_2$ to CO and $CH_4$. The enhanced activity resulted from the formation of a p-n heterojunction between NiO and CN, while $Ni_2P$ governed the activation and adsorption of $CO_2$ reactant species. The catalytic proficiency of TMPs towards photochemical HER and $CO_2RR$ applications are tabulated in Table 1.

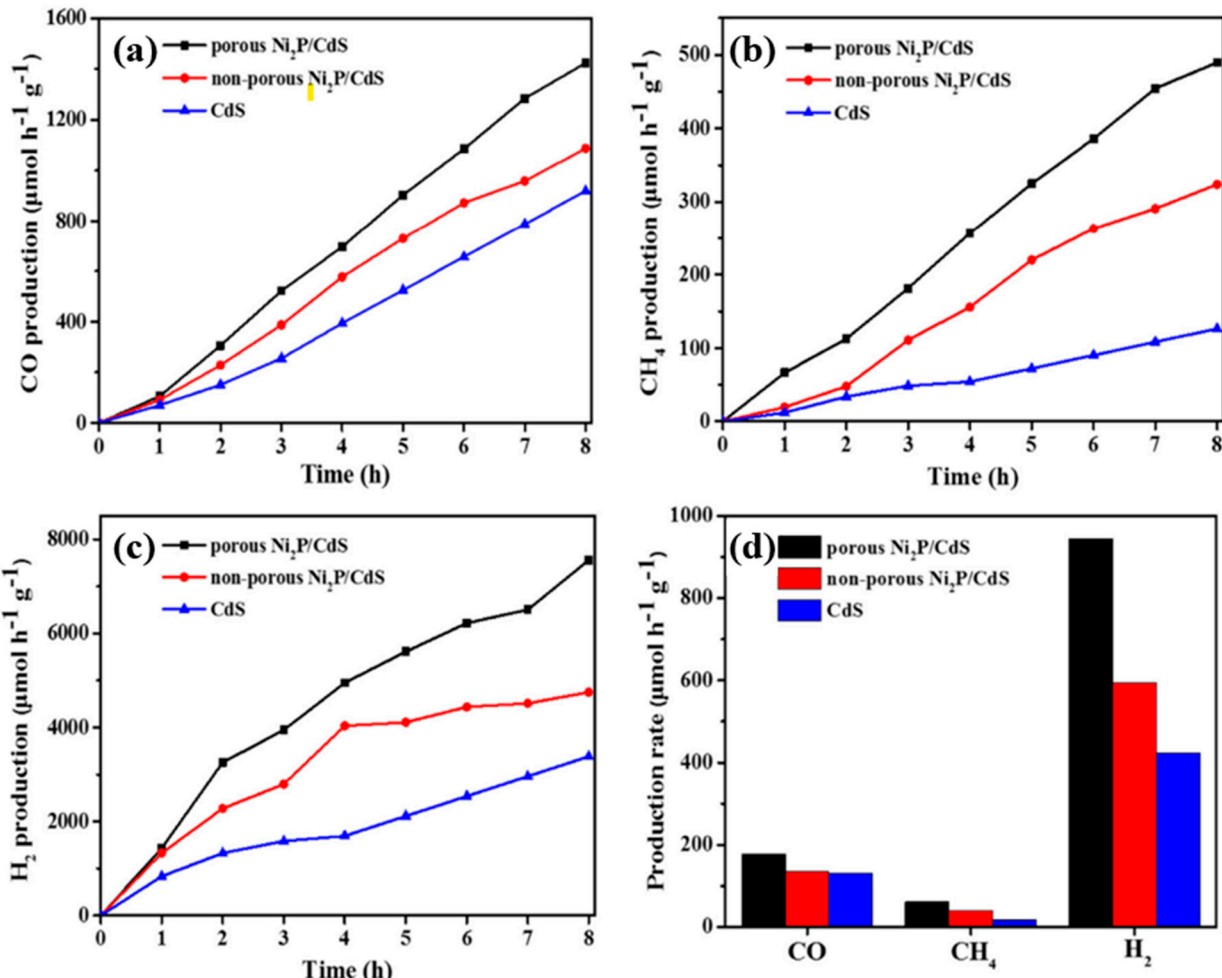

**Figure 8.** Photocatalytic rates of $CO_2$ reduction over porous and non-porous $Ni_2P/CdS$ composites: (**a**) CO, (**b**) $CH_4$, (**c**) $H_2$, (**d**) survey of various gases. [Reprinted with permission from Ref. [80]. Copyright 2023, Elsevier].

**Table 1.** Transition metal-phosphide-based photocatalysts for HER and $CO_2RR$ applications.

| Catalysts | Synthesis Method | Application | Band Gap (eV) | Rate of HER/$CO_2RR$ | Ref. |
|---|---|---|---|---|---|
| Ru-CoP-1:8/GCN | Chemical reduction | HER | 2.25 | 1172.5 $\mu$mol g$^{-1}$ h$^{-1}$ | [40] |
| $Ni_2P/NiO/CN$ | In situ gas–solid reaction | $CO_2RR$ | 2.6 | CO (1.506 $\mu$molg$^{-1}$ h$^{-1}$) $CH_4$ (0.29 $\mu$molg$^{-1}$ h$^{-1}$) | [72] |
| $Ni_2P/ZnIn_2S_4$ | Hydrothermal | HER | ~2 | 2066 $\mu$mol g$^{-1}$ h$^{-1}$ | [78] |
| $Cu_3P$-$Ni_2P$/g-$C_3N_4$ | Solvothermal | HER | 2.7 | 6529.8 $\mu$mol g$^{-1}$ h$^{-1}$ | [79] |
| $Ni_2P/CdS$ | Hydrothermal | HER and $CO_2RR$ | 2.16 | $H_2$ (111.3 mmol g$^{-1}$ h$^{-1}$) CO (178.0 $\mu$mol g$^{-1}$ h$^{-1}$) $CH_4$ (61.2 $\mu$mol g$^{-1}$ h$^{-1}$) | [80] |

**Table 1.** *Cont.*

| Catalysts | Synthesis Method | Application | Band Gap (eV) | Rate of HER/CO$_2$RR | Ref. |
|---|---|---|---|---|---|
| CoP/rGO | Hydrothermal | CO$_2$RR | - | CO (47,330 µmol g$^{-1}$ h$^{-1}$) | [81] |
| CoP/CNT | Hydrothermal | CO$_2$RR | - | CO (39,510 µmol g$^{-1}$ h$^{-1}$) | [81] |
| FeP/CN | Thermal decomposition | CO$_2$RR | 2.40 | CO (5.19 µmol g$^{-1}$ h$^{-1}$) | [83] |
| NiCoP/g-C$_3$N$_4$ | Thermal polymerization | HER | 2.69 | 5162 µmol g$^{-1}$ h$^{-1}$ | [85] |
| Ni-MoP@NP$_{PF}$ | Electrospinning | CO$_2$RR | 1.42 | CO (953.53 µmol g$^{-1}$ h$^{-1}$) | [86] |
| WP-NC/g-C$_3$N$_4$ | Facile sonication | CO$_2$RR | 2.8 | CO (376 µmol g$^{-1}$h$^{-1}$) | [93] |
| WP-NC/g-C$_3$N$_4$ | Thermal polymerization | HER | ~2.8 | 1217.6 µmol g$^{-1}$ h$^{-1}$ | [96] |
| Fe doped CoP | Self-assembly | CO$_2$RR | - | CO (21.0 µmol h$^{-1}$) | [97] |
| NiCoP nanosheets | Wet chemical and phosphorization | HER | - | 238.2 mmol g$^{-1}$ h$^{-1}$ | [98] |
| FeP/CdS | Solvothermal | HER | 2.32 | 37.92 mmol g$^{-1}$ h$^{-1}$ | [99] |

## 6. TMP Electrocatalysts for HER via Water Splitting

Over the last few years, a colossal number of attempts have been dedicated to the advancement of TMP-based electrocatalysts for propelling the applications of HER, and considerably accomplishments have been achieved by ameliorating the final results through heteroatom doping and nanostructure engineering [100–102]. Li et al. [74] synthesized Co$_2$P/Ni$_2$P nanohybrids for HER via water splitting. The as-developed nanohybrids required a very low overpotential (51 mV) to achieve 10 mA cm$^{-2}$ alongside noteworthy operational durabilities. This catalytic performance exceeded most of the reported non-noble TMP-based electrocatalysts. The exceptional results could be attributed to the large surface area, an abundance of active sites, and robust interfacial contact between Ni$_2$P and Co$_2$P. In another reported study, Cho et al. [75] examined the HER activity of Co-, Ni-, and Mn-doped FeP nanoparticles. The electrochemical results showed that Co-FeP nanoparticles required a 126 mV overpotential to attain 10 mA cm$^{-2}$ and exhibited higher cathodic current density than Ni- and Mn-doped FeP because of their fast charge transfer rate and high, electrochemically active surface area. In another study, Co-, Fe-, Mn-, Na-, Cr-, Li-, V-, Nb-, Ti-, Pb-, and Sn-doped Ni$_2$P electrocatalysts were investigated by Xiong et al. [76] for their HER catalytic activity. The electrochemical results demonstrated that Fe- and Co-doped Ni$_2$P exhibited superior performances similar to Pt-like activity with a very low overpotential of 31 mV at 10 mA cm$^{-2}$. These results are ascribed to the charge redistribution on the catalyst's surface, produced by the doping effect. Li et al. [103] fabricated MoP/MoNiP@C heterostructures to examine their electrochemical HER activity, and as a result, the as-prepared electrocatalyst showed remarkable performance, with a 134 mV overpotential and a 66 mV dec$^{-1}$ Tafel slope. These excellent results can be ascribed to the synergistic effect between MoNiP and MoP nanoparticles that augmented the active sites of the catalyst. The LSV (linear sweep voltammetric) curves and Tafel plots of as-prepared electrocatalysts towards HER are illustrated in Figure 9a,b, whereas cyclic voltammetric (CV) curves and a double-layer capacitance plot of MoP/MoNiP@C are demonstrated in Figure 9c,d. As shown in Figure 9e, the MoP/MoNiP@C impedance is relatively smaller than other electrocatalysts, which corroborates the higher catalytic performance and outstanding conductivity of MoP/MoNiP@C. To examine the robustness of MoP/MoNiP@C for longer runs, stability cycles of the LSV curves were recorded for 2000 cycles, and minor changes in the cathodic curve were observed, as shown in Figure 9f. The onset of Figure 9f depicts the time-dependent current density curve of the optimized electrocatalyst. The fabrication flowchart of MoP/MoNiP@C is illustrated in Figure 10.

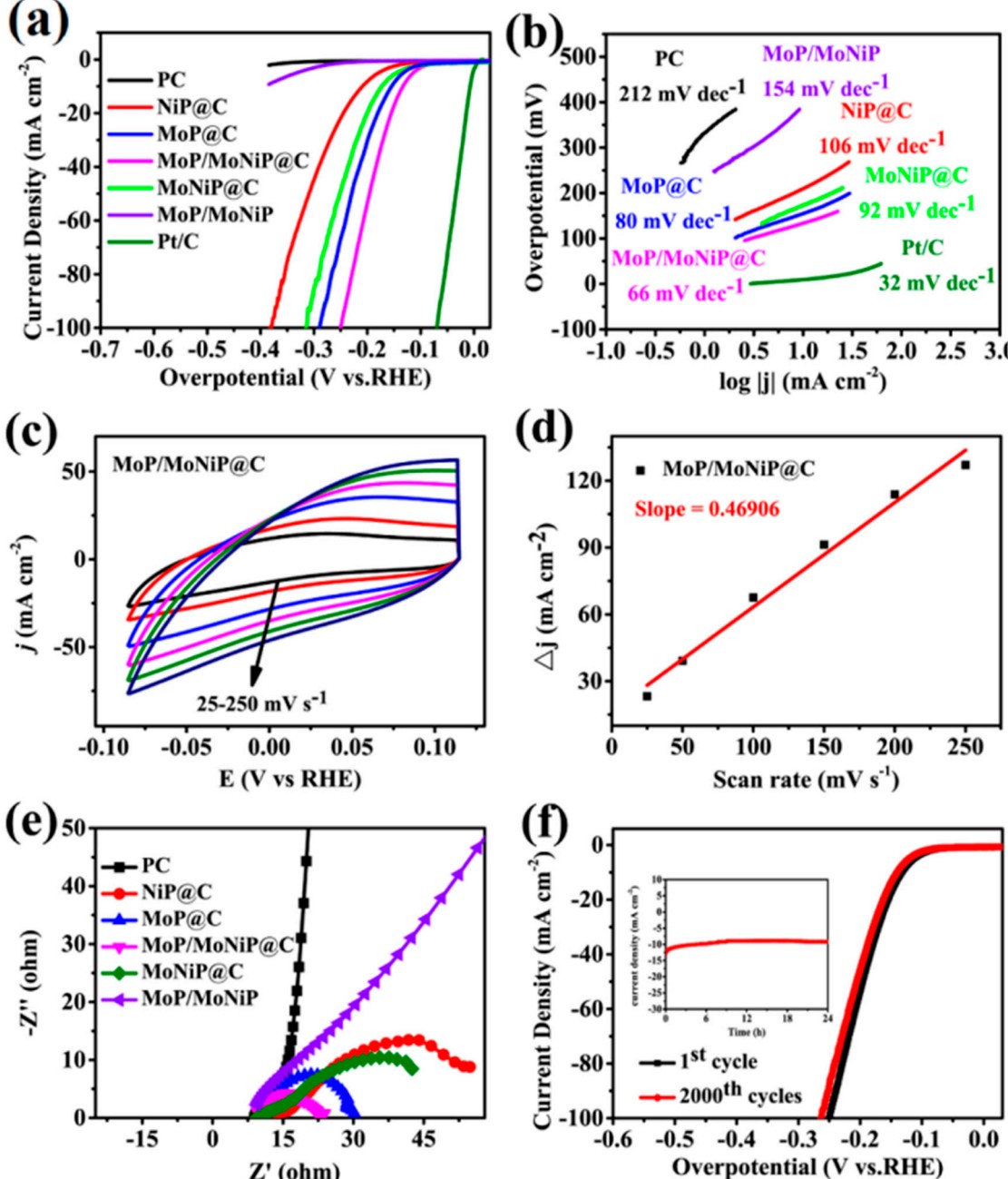

**Figure 9.** (**a**) LSVs of PC, NiP@C, MoP@C, MoNiP@C, MoP/MoNiP, MoP/MoNiP@C, and Pt/C in 0.5 M $H_2SO_4$ solution and (**b**) Tafel plots of PC, NiP@C, MoP@C, MoNiP@C, MoP/MoNiP, MoP/MoNiP@C, and Pt/C. (**c**) CV plots of MoP/MoNiP@C at a scan rates ranging from 25 to 250 mV $s^{-1}$ and (**d**) linear fitting of Δj vs. scan rates of MoP/MoNiP@C. (**e**) EIS spectra of PC, NiP@C, MoP@CMoNiP@C, MoP/MoNiP, and MoP/MoNiP@C. (**f**) LSVs of MoP/MoNiP@C in 0.5 M $H_2SO_4$ before and after 1000 cycles, and the inset indicates the time-dependent current density curve at an overpotential of 140 mV for 24 h. [Reprinted with permission from Ref. [103]. Copyright 2021, American Chemical Society].

**Figure 10.** Synthesis Process for MoP/MoNiP@C. [Reprinted with permission from Ref. [103]. Copyright 2021, American Chemical Society].

Yang et al. [104] designed $Ni_2P/MoO_2/NF$ nanorods-type heterostructured electrocatalyst for HER water splitting. As-prepared electrocatalysts exhibited very low overpotential (34 mV), which was attributed to the combined effect between $MoO_2$ and $Ni_2P$ that was enhanced due to the robust electronic coupling effect. In a reported work investigating the HER activity, Kang et al. [105] synthesized NiFeP@C electrocatalyst that showed notable results, including the low overpotential of 160 mV towards HER. The enhanced results accredited to the synergistic effect between P, Fe, Ni, and C, which accelerated the pace of charge transfer and escalated the electrocatalytic performance. Chen et al. [106] fabricated Ru-MnFeP/NF electrocatalysts to examine HER performance. The results showed a low overpotential of 35 mV alongside high stability for 50 h. Liu et al. [107] engineered Mn- and Ni-deposited FeP (Ni-Mn-FeP) electrocatalyst and investigated its HER activity. The as-prepared electrocatalyst demonstrated enhanced results with 103 mV ultralow overpotential. The combination technique of co-doped high-valence and low-valence metals motivated the advancement of high-activity and functional catalysts. In a reported study, Er-doped NiCoP/NF nanowires were fabricated by Zhang et al. [108] to assess their HER activity. The electrocatalytic efficiency of the as-prepared electrocatalyst was ascribed to its lowered overpotential value of 46 mV to reach 10 mA cm$^{-2}$ cathodic current density. The advanced performance of Er-doped NiCoP/NF was ascribed to the amalgamation of Er to NiCoP that considerably adjusted d-band centers alongside electronic structure of Co and Ni atoms by altering to lower energies with regards to Fermi level and also enhanced the HER/OER intermediate Gibbs free energies.

## 7. TMP Electrocatalysts for CO₂RR Application

TMPs have emerged as multi-facet materials with fascinating electronic and structural features, which have led to impeccable catalytic activity for critical energy transformation such as $CO_2$ reduction [109,110]. There are several recent reports of electrochemical CO₂RR applications of TMP-based nanocatalysts. Downes et al. [111] synthesized $Cu_3P$ nanoparticles to investigate their electrochemical CO₂RR performance. The results showed the formation of formate with as high as 8% Faradaic efficiency (FE). The enhanced activity and stability were accredited to using a solution-based molecular precursor method for $Cu_3P$ fabrication that offered multi-dimensional opportunities for altering the morphology, surface chemistry, and composition to attain notable $CO_2$ conversion efficiency.

In another work, Ji et al. [82] incorporated FeP on a Ti mesh (FeP/TM) that functioned as an effective electrocatalyst for $CO_2$ reduction to transform into alcohols with 94.3% $FE_{CH3OH+C2H5OH}$ and 80.2% $FE_{CH3OH}$. The improved results were attributed to the combining effect between two adjacent Fe atoms, as revealed by density functional theory (DFT) calculations. Sun et al. [112] designed MoP/In-PC for electrocatalytic $CO_2$ reduction. The current density and FE of the as-designed electrocatalyst reached 43.8 mA cm$^{-2}$ and 96.5%, respectively. These noteworthy results were accredited to the advantageous feature of nanosized MoP, the adsorption ability of the robust $CO_2$ and $CO_2$-intermediate, the high interfacial charge transfer, and the combining effect between In-doped carbon supports and MoP. Kim et al. [113] synthesized $Ni_2P/Ho_2O_3$ core–shell nanoparticles (CSNPs) to examine their scope in electrochemical $CO_2RR$ applications. As-synthesized electrocatalysts generated acetone with 25.4% FE that was attributed to the synergistic effect between $Ni_2P$ and $Ho_2O_3$. Figure 11 depicts the synthesis steps for designing $Ni_2P/Ho_2O_3$ catalytic systems. The four main value-added products in the form of $H_2$, HCOOH, $CH_3OH$, and $(CH_3)_2CO$ resulted from the reduction of $CO_2$, as revealed in the Figure 12a, $Ni_2P/Ho_2O_3$ core–shell nanoparticles (CSNPs) have been found to be very beneficial for the electrocatalytic behavior towards $(CH_3)_2CO$. While $Ho_2O_3$ nanodisks (NDs) produced $H_2$, HCOOH, and $CH_3OH$, only $H_2$ was generated by $Ni_2P$ in electrocatalytic $CO_2$ reduction towards $C_2$ or $C_3$ pathways, as shown in Figure 12b–d.

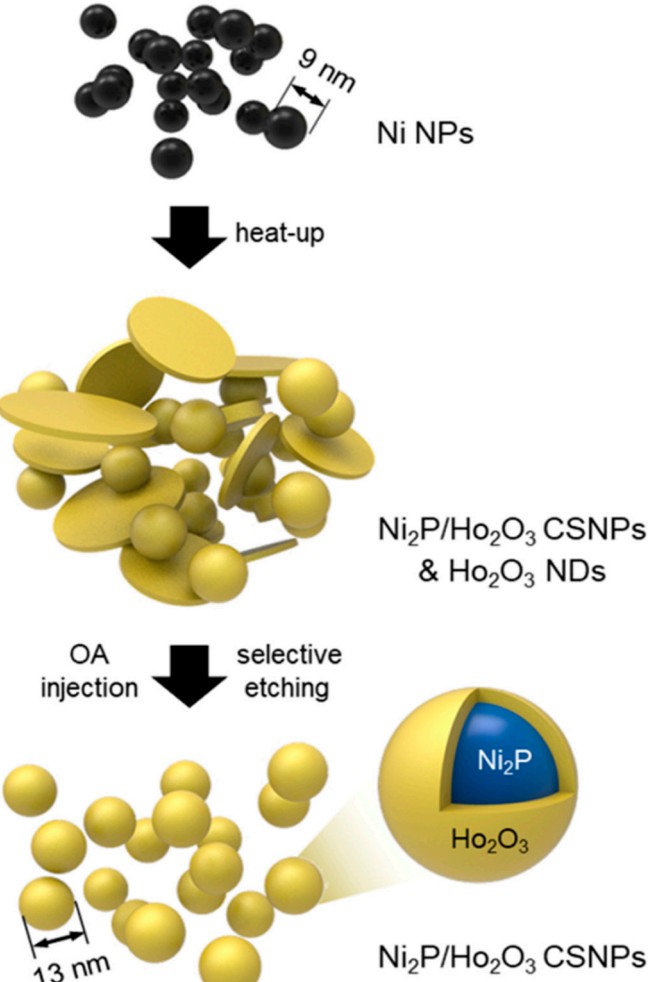

**Figure 11.** Schematic illustration of the experimental procedure for synthesis of C/A $Ni_2P/Ho_2O_3$ CSNPs (NPs, NDs, and CSNPs stand for nanoparticles, nanodisks, and core–shell nanoparticles, respectively). [Reprinted with permission from Ref. [113]. Copyright 2020, American Chemical Society].

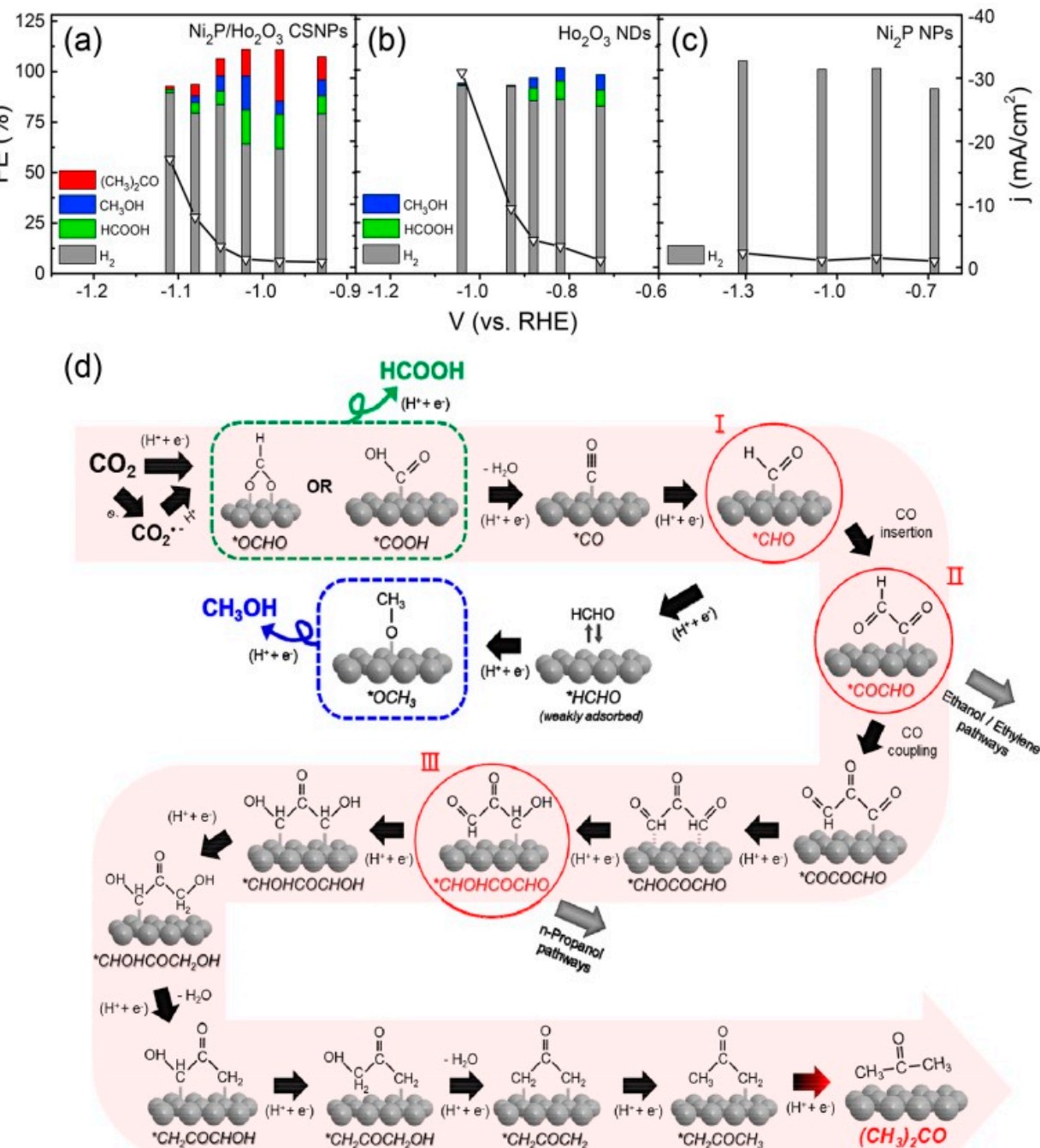

**Figure 12.** Potential-dependent FEs of $CO_2$RR products and total current densities obtained using (**a**) $Ni_2P/Ho_2O_3$ CSNPs, (**b**) $Ho_2O_3$ NDs, and (**c**) $Ni_2P$ NPs as electrocatalysts. (**d**) Suggested $CO_2$RR pathways toward $(CH_3)_2CO$. [Reprinted with permission from Ref. [113]. Copyright 2020, American Chemical Society].

Laursen et al. [114] fabricated $Cu_3P$ NS/Cu for electrocatalytic $CO_2$RR. The results showed the formation of formate with a 65 mV lower overpotential value and 0.9% FE. A $Fe_2P$ electrocatalyst was synthesized by Calvinho et al. [115] to examine the $CO_2$ conversion efficiency. The outcomes showed a 53% FE with ethylene glycol ($C_2$) product formation after $CO_2$ reduction, which inferred successful C-C coupling during the surface adsorption reaction mechanism. Banerjee et al. [116] fabricated $Ni_2P$ electrocatalyst for the reduction of $CO_2$, and the result demonstrated the formation of formaldehyde along with the surface hydrogen affinity and dynamic reconstruction of the surface through adsorption of H

that promoted the C–C coupling and selective reduction of $CO_2$ on $Ni_2P$. The catalytic proficiency of TMPs towards electrochemical HER and $CO_2RR$ applications are tabulated in Tables 2 and 3, respectively.

**Table 2.** Transition-metal-phosphide-based electrocatalysts for HER application.

| Electrocatalyst | Synthesis Method | Application | Overpotential (mV) | Tafel Slope (mV dec$^{-1}$) | Ref. |
|---|---|---|---|---|---|
| $Co_2P/Ni_2P$ | Thermal phosphorization | HER | 51 | - | [74] |
| Co-FeP | Phosphorization | HER | 126 | 63.6 | [75] |
| Co-Ni$_2$P | Synthetic method | HER | 31 | 47 | [76] |
| MoP/MoNiP@C | Calcination and phosphorization | HER | 134 | 66 | [103] |
| Ni$_2$P/MoO$_2$/NF | Phosphorization | HER | 34 | 45.8 | [104] |
| NiFeP@C | Calcination | HER | 160 | 75.8 | [105] |
| Ru-MnFeP/NF | Phosphorization | HER | 35 | 69 | [106] |
| Ni-Mn-FeP | Phosphorization | HER | 103 | - | [107] |
| Er-NiCoP/NF | Phosphorization | HER | 46 | - | [108] |

**Table 3.** Transition-metal-phosphide-based electrocatalysts for $CO_2RR$ application.

| Electrocatalysts | Synthesis Method | Power Density/ Current Density | Faradaic Efficiency | Ref. |
|---|---|---|---|---|
| Cu$_3$P/C | Hydrothermal | 2.6 mW cm$^{-2}$ | 47% (CO) | [28] |
| AgP$_2$ | Self-assembly | $-8.7$ mA cm$^{-2}$ | 82.0% (CO) | [63] |
| FeP/TM | Hydrothermal | - | 94.3% (CH$_3$OH + C$_2$H$_5$OH) | [82] |
| Cu$_3$P | Thermal decomposition | - | 8% (Formate) | [111] |
| MoP@In-PC | Solid state | 43.8 mA cm$^{-2}$ | 96.5% (HCOOH) | [112] |
| Ni$_2$P/Ho$_2$O$_3$ | Phosphorization | 0.95 mA cm$^{-2}$ | 25.4% (Acetone) | [113] |
| Cu$_3$P NS/Cu | Self-assembly | - | 1.1% $\pm$ 0.6% (Formic acid) | [114] |
| TiO$_2$/MnP | Annealing | - | 67% $\pm$ 5% (CO) 12.4% $\pm$ 1.4% (H$_2$) | [117] |

     The aforementioned path-breaking reports are proof for the ongoing research on TMPs as cocatalyst or electrocatalyst in HER via water splitting and $CO_2RR$ applications. However, there are still many mysteries unsolved regarding the synthesis of TMPs that comprise tail gas post-treatment and unstable reactants, which create obstacles for the bright future of TMPs as electrocatalysts. Therefore, coping with these bottlenecks of TMP nanostructures can simplify their success in energy-driven catalytic applications at scalable points.

## 8. Future Perspectives

     The outlook of nanocatalysis-assisted HER and $CO_2RR$ is propitious thanks to wide array of physicochemical modulations that can be carried out in different nanomaterials. Alleviating the cost of green HER and $CO_2RR$ applications is fundamental for applying these sustainable pathways to a scalable utility in day-to-day life. Therefore, designing cheap catalytic systems as an economical substitute for noble metal catalysts is the need

of the hour in order to cope up with the latter's whopping costs, which currently hinders the widespread exploitation of green HER and $CO_2RR$ applications [117–120]. In the last few years, researchers have realized the potential of TMPs as tremendously efficient catalytic systems able to address the energy related applications at scalable points [121–124]. Extensive energy-focused research operations have been performed by employing TMP nanostructured catalytic systems [125]. TMPs are emerging advanced functional materials that have recently exhibited great potential in catalyzing HER and $CO_2RR$ to generate value-added chemicals and fuels [126–128]. On account of their versatile physicochemical properties, TMPs are beneficial in a broad range of applications, varying from catalysis to energy storage devices. TMPs come forth as remarkable catalytic systems which offer a high range of performances and great robustness. Their inexpensiveness and recent ground-breaking advances promise to fill the void between research and commercial applicability.

Experimental and computational investigations of TMP nanostructures highlight their marvelous catalytic efficiency, stability, and selectivity towards HER and $CO_2RR$ applications [129]. However, the extent of their catalysis hinges upon exposed active sites, which govern the surface adsorption reactions. Therefore, it is fundamentally important to enhance the interaction between TMPs' surface sites and the reactant molecules to escalate the turnover frequency of generating $H_2$ and another chemical feedstock [130]. One major bottleneck of TMP-assisted nanocatalysis is the scarcity of reports which systematically explore and elucidate the reaction mechanism behind their inherent catalytic proficiency at an atomic scale. To examine the in-situ behavior of TMPs, sophisticated characterization techniques such as X-ray absorption and operando and in situ Raman spectroscopies can be upgraded in accordance with the reaction conditions for probing the intermediate steps involved during HER and $CO_2RR$ applications. In addition, theoretical models should be developed to determine the thermodynamic landscape and kinetics of surface reactions on the active TMPs centers.

## 9. Conclusions

Herein, we have discussed the exigency of renewable energy-driven $H_2$ production and $CO_2$ sequestration in order to meet the current energy demand through clean energy resources. TMP nanocatalysts have been reviewed as heterogeneous catalytic systems for HER and $CO_2RR$ applications. The structural advantages of TMPs have been discussed precisely, as structural engineering is one of the most sought-after pathways for enhancing their activity and stability. To understand the structural significance of TMPs, the role of P and M/P ratio was reviewed. We have summarized the most utilized synthetic methodologies for designing TMP nanostructures. Photocatalytic HER and $CO_2RR$ applications were surveyed collectively under the enlightenment of recent achievements of TMPs. Bimetallic NiCoP exhibits an enormous rate of $H_2$ production (238.2 mmol $g^{-1}$ $h^{-1}$), and heterostructured CoP/rGO was found to have a higher rate of reducing $CO_2$ to CO (47.33 mmol $g^{-1}$ $h^{-1}$). Correspondingly, recent advances in electrocatalytic HER and $CO_2RR$ applications were also reviewed, which corroborate the diverse applicability of TMP nanocatalysts in energy-driven applications. Pristine $Ni_2P$ has emerged as a resourceful electrocatalyst for HER. Nevertheless, there are only a handful reports for electrocatalytic $CO_2$ reduction, which is at the cutting edge of technology and needs to be developed further in future environmental aspirations. For $CO_2RR$ applications, TMPs can be exploited in either gas-phase or liquid-phase reactions, which facilitate the conversion of $CO_2$ into chemical feedstocks such as $C_1$ and $C_2$ products and other higher hydrocarbons. The P content and physicochemical properties of TMPs play major roles in regulating their catalytic responses. Therefore, optimizing reaction conditions and synthetic routes is instrumental in developing these emerging functional materials for energy-related applications. TMPs can be a game-changer in the field of nanocatalysis thanks to their Earth-abundance and cost-effectiveness.

**Author Contributions:** S.S. is responsible for the preparation of the rough manuscript by surveying different reports and literature. S.A.A. collected the data and statistics from recent reports. U.F.M. is responsible for compiling the rough draft into manuscript form. I.S. refined the manuscript. T.A. is responsible for the conceptualization, supervision, accruing funding, analysis, and finalization of the manuscript. All authors have read and agreed to the published version of the manuscript.

**Funding:** T.A. thanks UGC, New Delhi, Government of India, for the Research Grant for In-service Faculty Members. The authors also thank SERB, CSIR, and MoE (SPARC/2018-2019/P843/SL) for financial support to Nano Chemistry and Nano Energy Labs.

**Data Availability Statement:** Not applicable.

**Acknowledgments:** T.A. thanks the various research schemes sponsored by SERB, BRNS, CSIR, and MoE-SPARC, Government of India, for the financial support to Nano Chemistry and Nano Energy Labs at Jamia Millia Islamia. T.A. also thanks UGC, New Delhi, for the Research Grant for In-service Faculty Members. S.S., S.A.A. and I.S. thank UGC, New Delhi, for the Research Fellowships.

**Conflicts of Interest:** The authors declare no conflict of interest.

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
