# Peer review of "Recent Advances in Transition Metal Phosphide Nanocatalysts for H2 Evolution and CO2 Reduction"

_catalysts, doi:10.3390/catal13071046_

Round 1

Reviewer 1 Report

In this review, the authors discuss the transition metal phosphides nanocatalysts for H2 evolution and CO2 reduction, considering their structural properties and the synthetic advancements in this regard. Transition metal phosphides are a class of materials that have been extensively studied towards energy application. Therefore, a review of this topic seems timely. However, the present review has some drawbacks, which I partially mention in the following. 

First of all, the review does not discuss the intrinsic properties of these materials, which allow better photocatalytic energy harvesting/conversion. Therefore, I suggest the authors elaborate on this point by including relevant references. In this regard, their physical/chemical properties and the role of synthetic methods in engineering their properties should be included. The discussions mainly follow the statements from referenced works, but the authors themselves don't have their own discussions. Indeed, it is the main point of any review article. I think this point also must be addressed during the revision. In order to emphasize the capability of these materials, I suggest the authors compare them with other materials such as metal oxides (EnergyTechnol.2018,6, 459 –469), metal sulfides (Catalysts 202212(11), 1316), etc., used for this purpose. Most of the photocatalytic reactions emphasized throughout the review contain some side products, which should be emphasized, particularly when discussing the green approach for energy harvesting or photocatalytic conversion. I think all these general comments will certainly help the authors to improve their manuscript. In addition to these, the manuscript contains many grammatical errors, typos, and poor sentence constrictions, that must be addressed during the revision.

The manuscript contains many grammatical errors, typos, and poor sentence constriction. I hope the authors can solve all these problems during the revision. 

Author Response

Response to Reviewer’s Comments (Reviewer #1):

First of all, the review does not discuss the intrinsic properties of these materials, which allow better photocatalytic energy harvesting/conversion. Therefore, I suggest the authors elaborate on this point by including relevant references. In this regard, their physical/chemical properties and the role of synthetic methods in engineering their properties should be included. The discussions mainly follow the statements from referenced works, but the authors themselves don't have their own discussions. Indeed, it is the main point of any review article. I think this point also must be addressed during the revision. In order to emphasize the capability of these materials, I suggest the authors compare them with other materials such as metal oxides (EnergyTechnol 2018,6, 459-469), metal sulfides (Catalysts 2022, 12(11), 1316), etc., used for this purpose. Most of the photocatalytic reactions emphasized throughout the review contain some side products, which should be emphasized, particularly when discussing the green approach for energy harvesting or photocatalytic conversion. I think all these general comments will certainly help the authors to improve their manuscript. In addition to these, the manuscript contains many grammatical errors, typos, and poor sentence constrictions, that must be addressed during the revision.

Answer: We appreciate the important suggestions of the learned reviewer for the betterment of this manuscript. On that account, we have incorporated the intrinsic properties of TMPs precisely which highlighted that how this class of materials enhance the photocatalytic activity. Addressing the second query of reviewer, we have included the key role of synthetic methods that are responsible for altering physical and chemical properties of TMP based materials. Furthermore, we have discussed the principle factors which dominate the photocatalytic and electrocatalytic HER and CO2RR results as per our understanding. As suggested by the learned reviewer, we have compared the TMPs with metal oxides and metal sulfides in the light of advised research articles. Moreover, as far as side products are concerned, they are the drawbacks of photocatalytic reactions of CO2RR and HER, our main aim is to focus on primary reaction which has been thoroughly discussed however we have mentioned about this drawback in the revised manuscript also. In the end, we have improvised the grammatical errors and language of the main manuscript as suggested by the learned reviewer. We hope this satisfies all the queries of reviewer.

Reviewer 2 Report

The aim of this study was the Recent Advances on Transition Metal Phosphides Nanocatalysts for H2 Evolution and CO2 Reduction. It was well designed research and was done in sustainable and acceptable level. In my opinion, this work provides some interesting results but there are some comments that I think they are able to improve the quality of this investigation. Accordingly, I would like to recommend it for publication after minor revision.

1) It seems that by the addition of more numerical data and some classifications in the abstract section, you will able to illustrate the research novelty, more clearly. Add more numerical results and some classifications to the abstract section.

2)  In page 3, it was written that “Transition metal phosphides (TMPs) are the emerging class of advanced materials that are believed to be potent candidates which are able to replace precious noble metal-based catalysts”. It seems that, comparing the results of TMPs and noble metal-based catalysts with each other and indication of results in the form of some figures or graphs or tables lead to the more clearly illustration of the high potential of the TMPs. 

3)   Impact of synthesis methods. It is obvious that the synthesis methods is one of the important factors which influence catalytic performance. If it is possible for you, add a paragraph or section to the manuscript and in this new section, compare the impact of various synthesis method of TMPs on the catalytic performance. You can also, evaluate the effect of the parameters which are important in a specific synthesis method and add the results to the above-noted section.

4) As cited in abstract section, you will able to illustrate the research novelty, more clearly by addition of more numerical data and some classifications. Therefore, add numerical results and some classifications to the conclusion section.

Moderate editing of English language required

Author Response

Response to Reviewer’s Comments (Reviewer #2):

Q1: It seems that by the addition of more numerical data and some classifications in the abstract section, you will able to illustrate the research novelty, more clearly. Add more numerical results and some classifications to the abstract section.

Answer: We highly acknowledge the important remark of the learned reviewer. On that basis, we have added the numerical data and classifications of TMPs based catalysts in the abstract section to provide more clarity about the scope of TMPs in HER and CO2RR applications.

Q2: In page 3, it was written that “Transition metal phosphides (TMPs) are the emerging class of advanced materials that are believed to be potent candidates which are able to replace precious noble metal-based catalysts”. It seems that, comparing the results of TMPs and noble metal-based catalysts with each other and indication of results in the form of some figures or graphs or tables lead to the more clearly illustration of the high potential of the TMPs. 

Answer: We appreciate the important suggestion of the skilled reviewer. We have illustrated a diagrammatical comparison of TMPs and noble metal along with an experimental report in the revised manuscript.

Q3:  Impact of synthesis methods. It is obvious that the synthesis methods is one of the important factors which influence catalytic performance. If it is possible for you, add a paragraph or section to the manuscript and in this new section, compare the impact of various synthesis method of TMPs on the catalytic performance. You can also, evaluate the effect of the parameters which are important in a specific synthesis method and add the results to the above-noted section.

Answer: We highly acknowledge the important comment of the reviewer. As per suggestion, we have added a paragraph in the revised manuscript which emphasizes the comparative analysis between different synthetic routes of TMPs and showcase their impact on the catalytic performance of TMPs based catalytic systems.

Q4: As cited in abstract section, you will able to illustrate the research novelty, more clearly by addition of more numerical data and some classifications. Therefore, add numerical results and some classifications to the conclusion section.

Answer: We do appreciate the suggestive approach of reviewer to enrich our literature survey. Based on that we have improvised conclusion section by adding numerical data and classifications in the revised manuscript.

Reviewer 3 Report

This review mainly reports recent advances on transition metal phosphides nanocatalysts for H2 evolution and CO2 reduction. It is a topic of interest to the researchers in the related areas. However, there are some errors which are needed to be corrected in the paper.

1. The horizontal axis names of a, c, f in Figure 8 and a, b, and c in Figure 11 are not uniform. It will be better to understand when changing them to Overpotential (V vs. RHE).

2. How is it calculated that the FE corresponding to some potentials in a, b and c in Figure 11 exceeds 100%? What is the formula for calculating FE? Is it maybe a drawing error or other reasons?

3. In the view of author, the content of P in TMPs plays the central role in determining the catalysts’ catalytic activity. Based on that, it can be classified as transition metal rich TMPs (M-rich TMPs) and P atom rich TMPs (P-rich TMPs) depending on the M/P ratio. So, which type of catalyst performs better in the same catalytic reaction? In addition, what areas of catalytic applications are these two types of catalysts suitable for?

4. Some words do not have spaces between them, which may cause misunderstanding or difficult to understand. For example, "prismaticstructure" in the sixth row from the bottom of page 4, "ofM" and "ofTMPs" in the eighth and thirteenth rows from the bottom of page 4. Some similar errors in other places are not listed, and it is recommended to carefully check the words and sentences of the article.

5. A curious question: Has the author considered where the advantage of P is reflected compared to N, S and other non-metal element doping?

The overall writing of this review article is well thought out.

Author Response

Response to Reviewer’s Comments (Reviewer #3):

Q1: The horizontal axis names of a, c, f in Figure 8 and a, b, and c in Figure 11 are not uniform. It will be better to understand when changing them to Overpotential (V vs. RHE).

Answer: We appreciate the keen observations of the learned reviewer. Based on that, we want to state that since these are the copyright figures, therefore we do not have the data of obtained results.

Q2: How is it calculated that the FE corresponding to some potentials in a, b and c in Figure 11 exceeds 100%? What is the formula for calculating FE? Is it maybe a drawing error or other reasons?

Answer: We appreciate the query of the learned reviewer. Figure 11 is a copyright image from (10.1021/acs.inorgchem.0c03110), The colour represents the amount of FE of four main chemicals i.e., H2, HCOOH, CH3OH, and (CH3)2CO, in (Figure 4a). It does not mean that overall, FE is exceeding 100%. I hope this satisfies the query of learned reviewer.

Q3: In the view of author, the content of P in TMPs plays the central role in determining the catalysts’ catalytic activity. Based on that, it can be classified as transition metal rich TMPs (M-rich TMPs) and P atom rich TMPs (P-rich TMPs) depending on the M/P ratio. So, which type of catalyst performs better in the same catalytic reaction? In addition, what areas of catalytic applications are these two types of catalysts suitable for?

Answer: We highly acknowledge the important comment of the reviewer. As per the results of metal rich TMPs (M-rich TMPs) and P atom rich TMPs (P-rich TMPs) catalytic systems, it has been found that M-rich TMPs catalysts displayed better performances towards HER and CO2RR applications due to the metal-metal and metal-phosphorus bonds that can endow TMPs with better electron conductivity and thermal stability. Also, the results of both the classes of TMPs catalysts suggest the potential of this class towards sustainable energy related applications. 

Q4: Some words do not have spaces between them, which may cause misunderstanding or difficult to understand. For example, "prismaticstructure" in the sixth row from the bottom of page 4, "ofM" and "ofTMPs" in the eighth and thirteenth rows from the bottom of page 4. Some similar errors in other places are not listed, and it is recommended to carefully check the words and sentences of the article.

Answer: We appreciate the reviewer’s keen observations and we have modified it accordingly in the revised manuscript.

Q5:  A curious question: Has the author considered where the advantage of P is reflected compared to N, S and other non-metal element doping?

Answer: We really appreciate the metaphysical query of learned reviewer for the cognition about our thought-process for considering P based catalysts as compared to other non-oxide classes of nitrides and sulfides. On that basis, we want to state that TMPs based catalysts have recently attracted the interest of researchers for sustainable energy related applications due to their remarkable physicochemical properties that have been discussed in this manuscript. However, we have not directly compared the performance of TMPs catalysts to their nitrides and sulfides counterparts. For that, we really acknowledge the important suggestion of the reviewer, which we will surely embrace in our future review articles centered upon non-oxide based catalysts. We hope this satisfies the query of skilled reviewer.

Round 2

Reviewer 1 Report

The authors have significantly improved the manuscript. I suggest it be accepted for publication in Catalysts.